# Predicting mean ribosome load for 5'UTR of any length using deep learning

**Alexander Karollus** [1], **Žiga Avsec** [1,2], **Julien Gagneur** [1,3,4] *

**1** Department of Informatics, Technical University of Munich, Garching, Germany, **2** Graduate School of Quantitative Biosciences (QBM), Ludwig-Maximilians-Universität München, Munich, Germany, **3** Institute of Human Genetics, Technical University of Munich, Munich, Germany, **4** Institute of Computational Biology, Helmholtz Zentrum München, Neuherberg, Germany

* gagneur@in.tum.de

## Abstract

The 5' untranslated region plays a key role in regulating mRNA translation and consequently protein abundance. Therefore, accurate modeling of 5'UTR regulatory sequences shall provide insights into translational control mechanisms and help interpret genetic variants. Recently, a model was trained on a massively parallel reporter assay to predict mean ribosome load (MRL)—a proxy for translation rate—directly from 5'UTR sequence with a high degree of accuracy. However, this model is restricted to sequence lengths investigated in the reporter assay and therefore cannot be applied to the majority of human sequences without a substantial loss of information. Here, we introduced frame pooling, a novel neural network operation that enabled the development of an MRL prediction model for 5'UTRs of any length. Our model shows state-of-the-art performance on fixed length randomized sequences, while offering better generalization performance on longer sequences and on a variety of translation-related genome-wide datasets. Variant interpretation is demonstrated on a 5'UTR variant of the gene HBB associated with beta-thalassemia. Frame pooling could find applications in other bioinformatics predictive tasks. Moreover, our model, released open source, could help pinpoint pathogenic genetic variants.

## Author summary

The human genome carries a complex code. It consists of genes, which provide blueprints to assemble proteins, and regulatory elements, which control when, where, and how often particular genes are transcribed and translated into protein. To read the genome correctly and specifically to find the causes of inherited diseases, we need to be able to find and interpret these regulatory elements. Here, we focus on particular regions of the genome, the so-called 5' untranslated regions, which play an important role in determining how often a transcribed gene is translated into protein. We develop deep learning models which can quantitatively interpret regulatory elements in human 5' untranslated regions and use this information to predict a proxy of the translation efficiency. Our model generalizes a previous model to 5' untranslated regions of any length, just as they are encountered in natural human genes. Because this model requires only the sequence as input, it can give estimates for the impact of mutations in the sequence, even if these particular

**Data Availability Statement:** All data necessary to replicate the results can be found under https://doi. org/10.5281/zenodo.3584237. The code can be found under https://github.com/Karollus/5UTR.

**Funding:** Ž.A. and J.G. were supported by the German Bundesministerium für Bildung und Forschung (BMBF) through the project MechML (01IS18053F). The funders had no role in study design, data collection and analysis, decision to publish, or preparation of the manuscript.

**Competing interests:** The authors have declared that no competing interests exist.

mutations are very rare or entirely novel. Such estimates could help pinpoint mutations that disrupt the normal functioning of gene regulation, which could be used to better diagnose patients suffering from rare genetic disorders.

This is a *PLOS Computational Biology* Methods paper.

## Introduction

Eukaryotic cells make use of complex regulatory mechanisms, which allow precise control of the conversion of genetic information into functional proteins. Understanding how these mechanisms are encoded in regulatory sequences is therefore essential to both understand how healthy cells function and which mutations can predispose them to disease. Much progress has been made in understanding the control of transcription. However, mRNA abundance, while very helpful, is often not sufficient to accurately predict protein abundance [1–5]. This reflects an important role for regulatory mechanisms that act after transcription, such as those controlling translation.

The 5' untranslated regions (UTR) of RNA transcripts plays a key role in translation [6]. According to the standard scanning model of translation initiation, the ribosome binds to the 5' cap of the mRNA and scans along the 5'UTR until it finds a suitable translation initiation site (TIS), at which point it will begin the process of protein assembly [7,8]. This process is generally described as leaky, as the ribosome can skip a TIS. The ribosome is more likely to skip weaker TIS, i.e. TIS with an unfavourable sequence context, as opposed to strong TIS [9]. A strong TIS will usually be composed of the AUG start codon flanked by a sequence similar to the Kozak consensus GCC(A/G)CC**AUG**G [10], although other features such as secondary structure will also play a role [11,12]. Once a TIS has been selected by the ribosome, it will continue translating until encountering an in-frame stop codon. Approximately 50% of human transcripts have a TIS and corresponding stop codon upstream of the canonical coding sequence, a structure commonly referred to as upstream open reading frame (uORF) [13].

The scanning process has important implications for the regulation of translation. For one, regulatory motifs can increase or reduce overall protein production by aiding or impeding the scanning ribosome. For instance, single nucleotide variants (SNV) affecting the Kozak sequence or uORFs have been shown to cause significant variation in protein abundance between humans, even if mRNA abundance is unaffected [14]. Differences in translation efficiency due to such variants have also been observed in a mouse hybrid system [15]. Additionally, mutations that introduce new TIS upstream of the canonical start codon (uTIS) can cause the ribosome to translate an altered protein. This may either correspond to a lengthened version of the canonical protein, or, if the new TIS is out of frame with respect to the canonical start, an entirely new, and likely dysfunctional, protein. As a result, variants in the 5'UTR can contribute to or even cause diseases and thus the analysis of such variation has clinical significance [13,16–18].

Much of the computational literature studying translation and the 5'UTR has focused on developing methods to classify whether a particular input sequence segment acts as a TIS or not [19–23]. Some studies also directly report which of several TIS in a sequence is most likely to be chosen [24], although in theory all tools in this category could be used for this purpose after slight modification. Generally, these computational methods provide accurate predictions for their chosen task. However, as their goal is to classify TIS they are not designed to provide a comprehensive estimate for the overall impact of a particular 5'UTR sequence on translation.

Recently a massively parallel reporter assay (MPRA) has been developed which provided a more complete quantification of the impact of the entire 5'UTR sequence on translation [25]. Specifically, more than 200,000 entirely random 5'UTR sequences were generated, each 50 nucleotides long, and fused with an enhanced green fluorescent protein coding sequence. Using the polysome profiling technique, the mean ribosome load (MRL), a metric of the average number of ribosomes associated to a given RNA and a proxy for translation efficiency, was measured for each sequence. This experimental setup thus allowed measuring the combined impact of a large variety of 5'UTR motifs on MRL, without any bias due to differences in the coding sequence or 3'UTR. The same experiment was additionally performed for a library of about 80,000 random 5'UTRs with lengths ranging from 25 to 100 nucleotides.

Using this data, convolutional neural network models predicting the MRL directly from the 5'UTR sequence were trained, including one model using the 50 nt long MPRA sequences and one model using the variable length MPRA data. Henceforth, they will be referred to as Optimus50 and Optimus100 respectively. These models are very accurate on their respective test sets and they are undoubtedly valuable tools to study the impact of different 5'UTR sequence features. However, as a result of the specific architecture used, the Optimus models learn position-specific weights (where the position is defined relative to the canonical start codon). As a result, neither model can yield predictions for sequences longer than the longest sequence in their training data. Longer 5'UTR sequences need to be truncated before they can be fed to the model, and thus any information contained in the truncated segments are lost. However, the average human 5'UTR contains about 200 nt, and thus many annotated human 5'UTR are significantly longer than 50 or 100 nucleotides [26]. As a result, Optimus MRL predictions may be incomplete and potentially unreliable for a large number of human transcripts. Moreover, the effects of variants disrupting motifs further than 100 nt from the canonical start cannot be quantified, making it difficult to apply the Optimus models to real human variant data. This is unfortunate because, as the authors have shown, the Optimus models do provide reasonable variant effect predictions for sequences which do not violate its length restrictions.

Here we develop a model which bridges this gap and extends the capabilities of the Optimus models to 5'UTR of any length. Such a model would then allow, in principle, to quantify the impact on MRL of any kind of variant, mutation or indel, anywhere in a 5'UTR, thus making the rich knowledge encoded in the MPRA data easily accessible to practitioners.

## Results

### Modelling 5'UTR of any length using frame pooling

To investigate the extent to which 5'UTR length varies across human transcripts, we computed the empirical cumulative distribution function of length for the 5'UTR annotated in GEN-CODE v19 (Fig 1A). It was found that only 30% of 5'UTR are 100 nt or shorter. Therefore, around 70% of human 5'UTR need to be truncated before they can be analysed by an Optimus model. If only Optimus50 is used, this number rises to 85%. Furthermore, as the median sequence has about 200 nt, it means that truncation will usually lead to a substantial loss of information. In the majority of cases, more than half of the sequence must be thrown away before an Optimus model can be used.

To create a model which provides mean ribosome load predictions for any 5'UTR, regardless of length, care must be taken to not introduce position-specific weights while nevertheless capturing and quantifying the regulatory motifs found in the sequence. One approach, which had been successfully used for this purpose in the past, is to combine convolutional layers with a global pooling operation ([27,28], reviewed in [29]). In such a setup, convolutional layers specialize on detecting the presence and strength of regulatory motifs, whereas the pooling

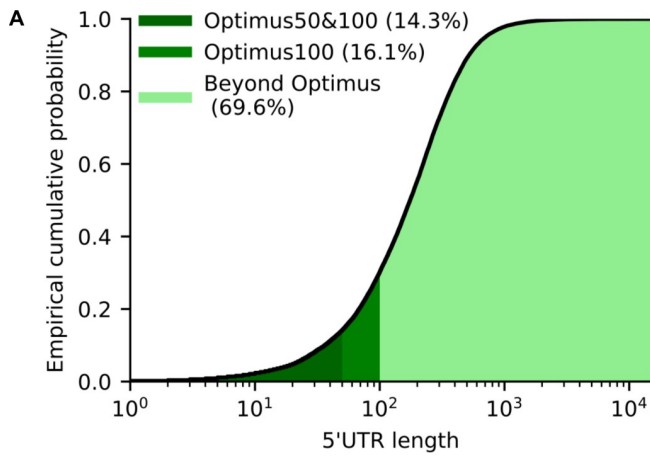

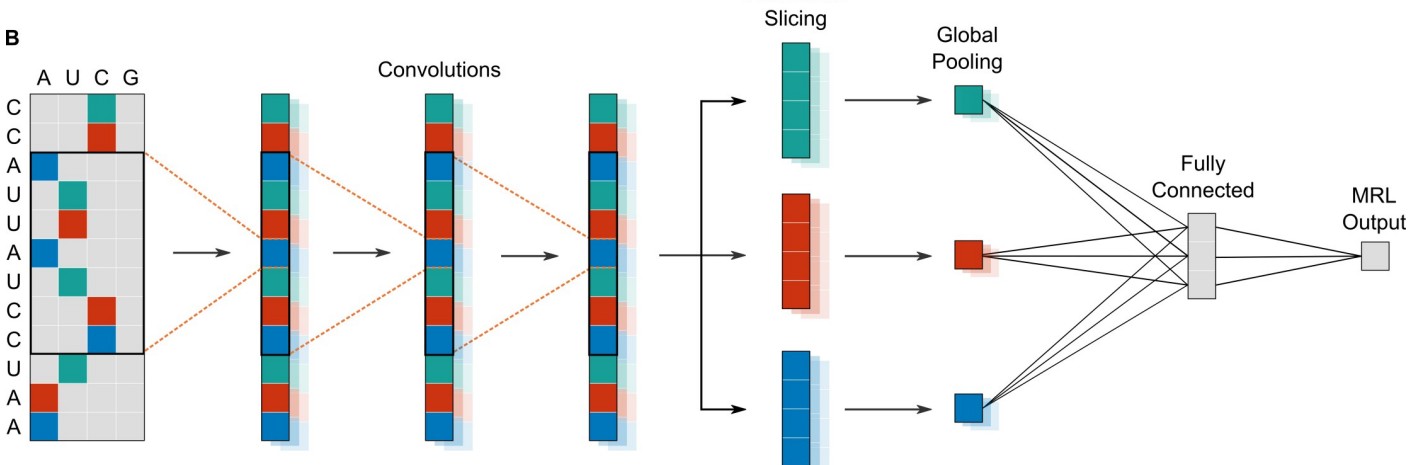

**Fig 1. Modelling 5'UTR of any length. A)** Empirical cumulative distribution of human 5'UTR lengths, according to GENCODE v19: only 14% (for Optimus50), or 30% (for Optimus100) of human 5'UTR sequences can be quantified by an Optimus model without information being lost due to truncation. **B)** Schematic of a frame pooling model: a one-hot-encoded 5'UTR is fed through 3 convolutional layers. For ease of visualization, the biological frame is indicated by colour. Then the convolution output is sliced according to the frame, and each frame is pooled separately, using global average and max pooling. The frame-specific information is then aggregated by a dense layer and a final linear layer assembles the mean ribosome load prediction.

layers aggregate this information across the sequence. However, as noted previously, the impact of upstream start codons (uAUG), depends heavily on whether they are located in-frame or out-of-frame with respect to the canonical start codon. As a result, simple global pooling will not generalize well to arbitrary length sequences since this operation loses the frame information.

To overcome this problem, we propose instead to first separate the convolutional output according to the underlying biological reading frame and then perform global pooling for each frame separately. This method ensures that the frame information is preserved and thus the network can differentiate between regulatory motifs located in-frame or out-of-frame with the coding sequence. We call this operation framewise pooling, or frame pooling for short.

The resulting model consists of three convolutional layers, followed by frame pooling, a fully connected layer and a linear layer to assemble the final MRL prediction (Fig 1B). Two global pooling operations are performed: global max pooling and global average pooling. Max pooling indicates whether a particular motif is strongly present in a particular frame, whereas average pooling roughly indicates how often a particular motif is present in each frame.

The model was trained three times: using the 50 nt MPRA sequences to allow comparison with Optimus50, using the 25–100 nt MPRA sequences to allow comparison with Optimus100 and using both datasets, to create a combined model. Since these datasets measure MRL on a slightly different scale, the combined model has an additional regression layer before the final output that learns a library-specific scaling. Henceforth the models will be referred to as FramePool50, FramePool100 and FramePoolCombined respectively.

These models have been integrated into the Kipoi API [30], allowing them to be applied with very little overhead to a VCF file containing human variant data. As a result, the models are easy to use and straightforward to integrate into existing variant annotation pipelines.

## Evaluating frame pooling on MPRA data

To ascertain that a model based on frame pooling still yields accurate predictions, despite no longer having detailed position information, it was tested on the same held-out test set as in the original Optimus study [25]. On these 20,000 sequences, the predictions of the frame pooling model (FramePool50) show a Pearson correlation of 0.964 with the observed MRL values, whereas Optimus50 had a correlation of 0.966 (Fig 2). Hence, despite having considerably fewer weights overall, and no position-specific weights in particular, FramePool50 still performs almost as well as Optimus50 on MPRA sequences. This demonstrates that frame pooling

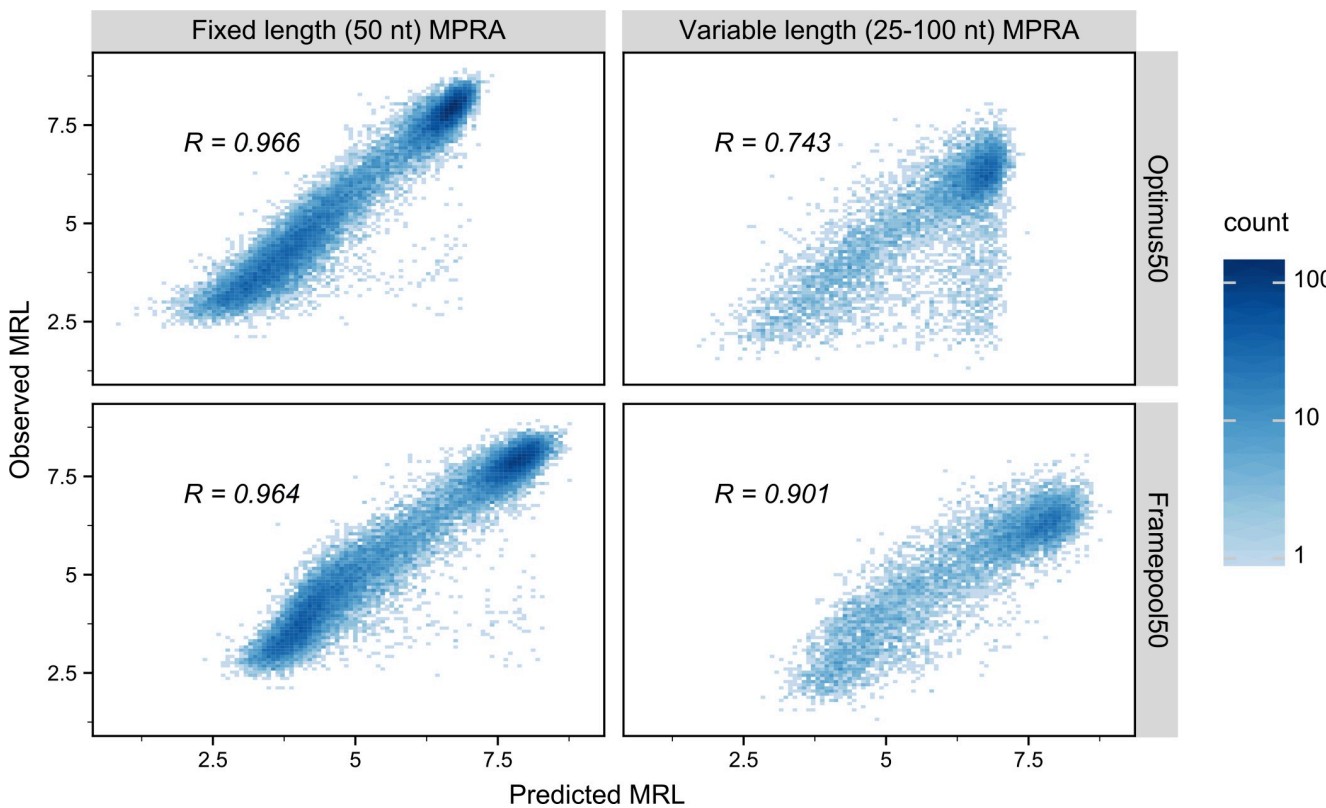

**Fig 2. Performance of Optimus50 and FramePool50 on MPRA test-sets.** Observed Mean ribosome load (MRL, y-axis) against predicted MRL (x-axis) for Optimus50 (top row) and FramePool50 (bottom row) on the 50 nt fixed-length MPRA dataset (left column) and the variable-length MPRA dataset (right column). Optimus50 performs very well when evaluated on a test set of 20,000 random 50 nt long 5'UTR sequences. However, its performance drops strongly when evaluated on a test set of 7,600 random 5'UTR sequences which vary in length from 25 to 100 nt. This is because sequences longer than 50 nt need to be truncated to fit the model. Despite not having position-specific weights, the frame pooling model captures almost as much signal as Optimus on the short MPRA sequences. Additionally, it generalizes very well to the variable-length MPRA data, strongly outperforming Optimus50. This shows the ability of the frame pooling approach to capture predictive signals even in sequences considerably longer than those it has been trained on.

is sufficient to capture almost all the signal present in MPRA data, with the distinct advantage of not needing to introduce constraints on the sequence length.

This advantage becomes apparent when applying the model to the other MPRA test set, consisting of 7,600 sequences, with lengths ranging from 25 to 100 nt (with 100 sequences for each length). Optimus50, due to its position-specific weights, cannot identify motifs located further than 50 nucleotides from the canonical start codon. As a result, it generalizes poorly to this variable-length test set, and the correlation drops to 0.743. FramePool50, despite being trained on the same data, does not face this restriction and generalizes much better, with a correlation of 0.901 (Fig 2). Note that this result compares well even with Optimus100, which was trained on the variable-length MPRA data, and has a correlation of 0.915 with the observed values of this test set (S1 Table). A similar pattern can be observed on (truncated) human sequences measured in the MPRA experiments (S1 Fig). Optimus50 and FramePool50 perform very similarly on human 5'UTR truncated to 50 nt (Pearson correlation: 0.889 vs 0.882), but frame pooling again generalizes better: for 25–100 nt sequences the correlation for Optimus50 drops to 0.7, whereas FramePool50 remains at 0.871. These results show that, despite its relative simplicity, the frame pooling operation allows for effectively generalizing the model to 5'UTR sequences considerably longer than those it has been trained on.

To ascertain that frame pooling represents an improvement over conventional global pooling strategies which also generalize to arbitrary length sequences, we additionally compared FramePool50 to models which i) use only global pooling, or ii) global pooling together with dilated convolutions. We find that FramePool50 offers considerably better generalization performance to variable length sequences than these alternatives (S4, S5 and S6 Figs). Furthermore, we also explored random forest models on 3-mer and 4-mer counts. Here we found that random forests supplied with explicit frame information (Methods, Random Forests) performed better than those without (S7 Fig), but neither could match the performance of FramePool50 (S8 Fig). Altogether these results indicate that the model structure encoded by the frame pooling operation is informative and not easily replicated by frame-unaware architectures.

## FramePool quantitatively predicts effects of uTIS motifs

To evaluate the predictive power of a frame pooling model on an independent data set, we applied it to large-scale perturbation assays probing upstream translation initiation site contexts. Noderer et al. [9] investigated every possible -6 to +5 context of a AUG site, whereas Diaz de Arce et al. [31] performed a similar analysis for every -3 to +4 context for selected alternative start codons, such as CUG and GUG. We evaluated the extent to which our model (FramePoolCombined, see S9 and S10 Figs for the results of this analysis for the other models) agrees with these estimates. This was achieved by injecting each of these uTIS motifs out-of-frame into a random sequence and then feeding them to the model twice: once with the central start codon "deactivated" (e.g. AUG replaced with AGG) and once in an "active form". The difference, in terms of fold change of the predicted MRL, between the active and inactive version was recorded (Fig 3A). The resulting scores were correlated with the relative strengths of the respective motifs (Fig 3B). We expect these correlations to be significantly negative, since stronger uTIS motifs should also lead to stronger reductions in predicted MRL if they are activated. For uAUG contexts we indeed measured a strong, negative Pearson correlation of -0.802. Hence, despite not having been trained explicitly for this task, the model has learnt to effectively distinguish between weak and strong uTIS motifs with high accuracy. For uCUG and especially uGUG, the correlations are weaker (-0.712 and -0.394 respectively) but still significant. Likely the training data was not sufficient to learn the full regulatory code of such

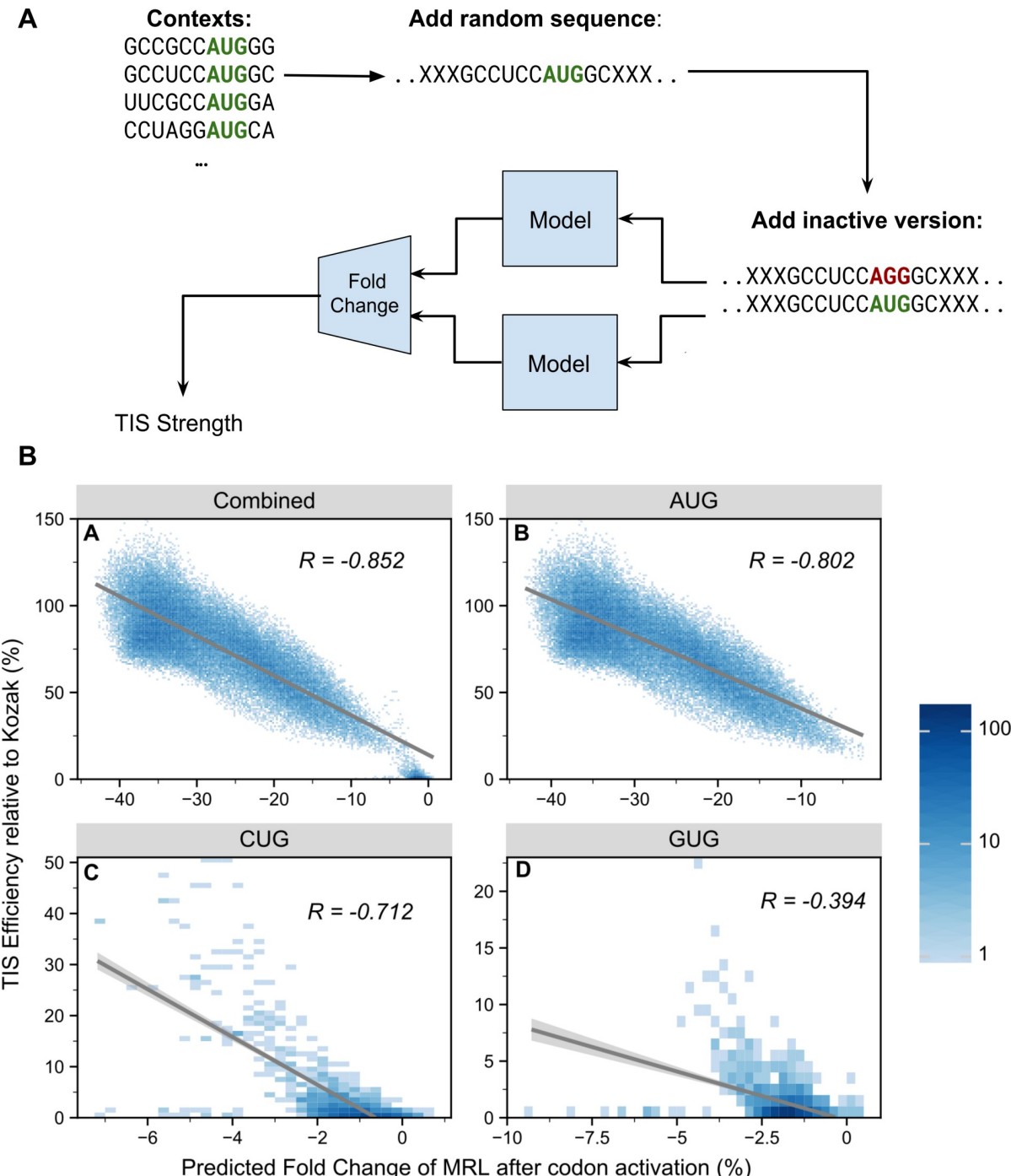

**Fig 3. Framepool predicts the strength of upstream transcription initiation sites. A)** The model was used to predict the effect of upstream TIS contexts by injecting them into random sequences and comparing predicted MRL fold change associated with upstream start codon activation (mutation AGG → AUG). **B)** These predictions (x-axis) were correlated with independent experimental measurements of the relative strength of these contexts (y-axis). Overall, a negative correlation is observed (top right) as expected because upstream TISs compete with canonical TISs. For AUG start codon data [9], the model correlates well with the strength estimates (top right, Pearson correlation of -0.802), with more efficient contexts causing a larger predicted effect on MRL. Negative correlations were also observed for the alternative start codons CUG (bottom left) and GUG (bottom right) data [31]. The correlations are weaker, particularly for GUG, likely reflecting the difficulty of detecting more subtle motifs in MPRA data. Nevertheless, the model has still learned to distinguish between strong and weak CUG/GUG motifs to some extent.

non-canonical TIS, as they generally are much weaker than uAUG, and thus more difficult to detect in MPRA data. Nevertheless, even here the model clearly makes some correct distinctions between stronger and weaker alternative start codon motifs. In summary, these results on independent perturbation data show that our model captured important components of the 5'UTR code regulating translation, and can be used to quantify the impact of variants on translation control.

## Evaluating frame pooling on endogenous genes

Having evaluated the performance of frame pooling on MPRA data, we next analysed whether it could also deliver meaningful predictions for endogenous genes.

Many experimental protocols exist to measure different aspects of translation efficiency for human transcripts, such as ribosome footprinting (Ribo-Seq, [32]) or using Mass-Spec to determine Protein-to-mRNA ratios (PTR, [33]) which all relate to mean ribosome load. Hence, such measures should correlate with the MRL predictions provided by our models. The correlations are expected to be lower though, because these measurements are not direct measurement of ribosome load and because in endogenous data, as compared to MPRA data, the coding sequence and the 3'UTR are not constants, but also vary between transcripts. Both of these features have been shown to considerably affect the ribosome load and the PTR [3,34]. Moreover, PTR is also dependent on protein degradation rates, which are not captured by MRL.

Nevertheless, we correlated the predictions of our models with such measures gathered from six different previously published studies. These include two Ribo-Seq studies performed on HEK293 cells [35,36] and one on PC3 cells [37]. In these three studies, RNA sequencing (RNA-Seq) and Ribo-Seq data are combined to compute a measure of translation efficiency comparable to ribosome load. Two studies measure PTR across human tissues [3,38]. As our model is not tissue specific, we used the median PTR across tissues. Lastly, Floor et al. [34] computed a mean ribosome load measure using a technique called Trip-Seq for all transcripts in HEK293 cells. This involves polysome profiling and is similar to the MRL measured in the MPRA experiments. Note that most of these studies compare several conditions but we only used data for control group cells.

For each of these datasets, the correlations were positive and statistically significant, albeit small, ranging from 0.11 to 0.25 (S2 and S3 Tables). Thus, despite the bias introduced by other sequence features and despite our models only being trained on purely random sequences, they nevertheless captured biologically relevant signals related to translational regulation.

Moreover, the MRL predictions of the frame pooling model correlated better with these various datasets than the Optimus predictions did (Fig 4). For example, on sequences longer than 50 nt, FramePool50 shows significantly better correlation on endogenous data than Optimus50 on 4 out of 6 datasets, with the last two showing no significant difference (Fig 4). Meanwhile, on sequences longer than 100 nt, FramePool100 significantly outperforms Optimus100 on two datasets, with the other four showing no significant difference. These results provide further evidence that frame pooling allows to generalize and provide meaningful predictions beyond the range of 5'UTR sequences used in training. Moreover, these results indicate that the ability to generalize to longer sequences is not restricted to synthetic MPRA sequences, but extends to human transcripts.

Eraslan et al. [3] identified a number of 5'UTR motifs which are not captured by our models, likely because their effects are too subtle or context-specific to be detected in MPRA data. A linear model which includes these motifs as additional predictors can explain more of the signal in PTR data than the MRL predictions of a frame pooling model alone, although the

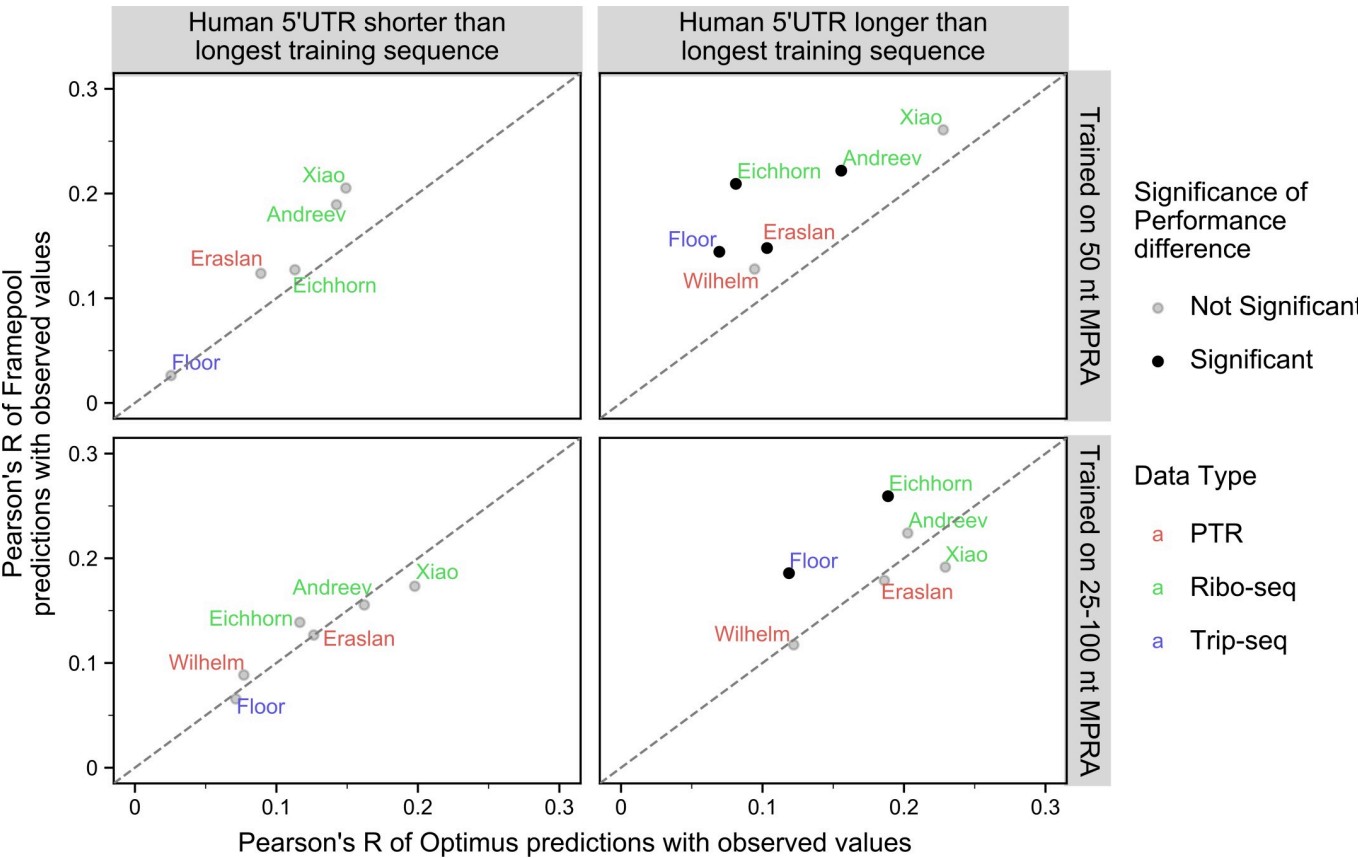

**Fig 4. Correlating Optimus and Framepool MRL predictions with a variety of translation-efficiency related measurements for human transcripts.**
Pearson's correlation between MRL predictions and various published translation-efficiency related measurements for Framepool (y-axis) against Optimus (x-axis) for human 5'UTR sequences shorter than the longest training sequence (left column) or larger (right column) and when the model is trained on 50 nt MPRA (top row) or the variable-length MPRA (bottom row). Dataset labels are colour-coded by measurement type: Protein-to-mRNA ratio (red), Ribo-Seq (green), and TripSeq (blue). For 5'UTR sequences shorter than 50 nt, no significant difference can be observed between FramePool50 and Optimus50 (the diagonal line indicates identical performance). For those longer than 50 nt, FramePool50 significantly outperforms Optimus on 4 out of 6 datasets. FramePool100 and Optimus100 show indistinguishable performance on 5'UTR sequences shorter than 100 nucleotides. For sequences longer than 100 nt, FramePool100 significantly outperforms Optimus on 2 out of 6 datasets, with inconclusive results for the other sets. This suggests that frame pooling captures additional signals from longer sequences in endogenous data. Significance is assessed by bootstrapping and corrected for multiple testing (Bonferroni, Methods).

difference is modest (using FramePoolCombined as the basis, the Pearson correlation rises from 0.171 to 0.199 when including the additional motifs). In further studies of translation efficiency, predictions from a frame pooling model could be used as an informative input feature to models trained on endogenous data.

## Quantifying the impact of variants

We next asked whether our models could identify functionally important nucleotides. We reasoned that single nucleotide variants predicted to have strong effects on ribosome loading should occur at locations which are more evolutionarily conserved, at least for functionally important genes. Hence, we predicted the effect of every single nucleotide variant on MRL throughout the 5'UTR of the canonical transcripts of genes strongly depleted for loss-of-function variants in the human population (LoF-intolerant genes, Methods). We observed that the stronger the effect FramePool100 predicted was, the more phylogenetically conserved the position was (PhyloP score [39] Methods, Fig 5). The trend was particularly pronounced within

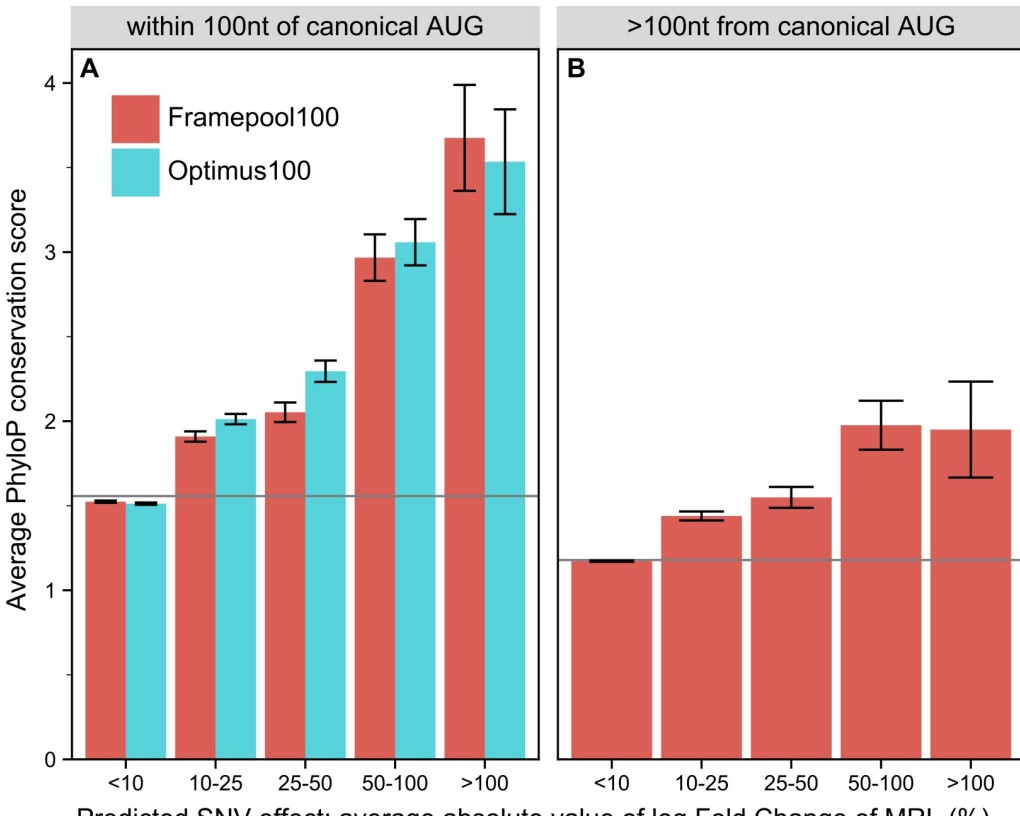

**Fig 5. Comparing the predicted impact of every possible 5'UTR SNV with conservation scores for loss-of-function intolerant genes. A)** For positions no further than 100 nt from the canonical transcript start codon, both Optimus100 and FramePool100 can predict the impact of SNV. Positions where a SNV is predicted to have a high impact on average also tend to be more strongly conserved. Error bars correspond to 95% confidence intervals. The grey line shows the average PhyloP score among all analysed 5'UTR position. **B)** For 5'UTR positions further than 100 nt from the canonical start, only FramePool100 can be applied. Again, positions where the SNV impact is predicted to be higher also tend to be more highly conserved. This further demonstrates that FramePool100 can identify biologically relevant regulatory elements even in sequences longer than those it has been trained on.

100 nt of the canonical start codon, whereby variants predicted to affect MRL by more than 50% showed an average PhyloP score roughly double than the average 5'UTR position (Fig 5A). The same trend was observed for Optimus100. Positions further than 100 nt from the canonical start could only be scored with FramePool100 and further exhibited a milder yet significant association between phylogenetic conservation and predicted SNV effect on MLR (Fig 5B). These results indicate that these models can identify positions in 5'UTR with important regulatory functions. Therefore, these models could be useful to flag variants which may have particularly deleterious impacts on translational control. In this regard, the frame pooling model has the advantage that it can deliver predictions for any 5'UTR variant, regardless how far it is located from the canonical start.

## Variant effect prediction and model interpretation

The FramePoolCombined model has been integrated into the model repository for genomics Kipoi [30]. This allows leveraging the diverse features of the Kipoi ecosystem. Most importantly, the FramePoolCombined model can be readily used for variant effect prediction including SNVs and indels from VCF files (Figs 6 and S11).

The Kipoi model reports variant effects in terms of $\log_2$ fold change of MRL. However, uTIS located out-of-frame will generally have a large impact on MRL, but in-frame uTIS may not necessarily change ribosome load much if they just lengthen the effective coding sequence. Nevertheless, such a change may still affect the function of the protein. To provide indication to practitioners whether such a protein lengthening has occurred, the Kipoi version of the model additionally reports two additional variant effect scores, representing the predicted variant effect after shifting the 5'UTR sequence by one or two frames. These artificial shifts trick the model into treating an in-frame variant as an out-of-frame variant. Variants creating in-frame uTIS and only lengthening the canonical protein typically have a small effect on MRL before shifting, and strong negative effects after shifting (S12 Fig).

To get an overview of what the model deems to be the most important motifs determining MRL in MPRA data, we used Tf-Modisco [40]. Tf-Modisco clusters segments of input sequences considered to be important to the respective predictions delivered by the neural network. The motifs found by Tf-Modisco correspond to known features of translational regulation, namely uAUG motifs with Kozak like flanking sequences and uORF stop codons (S13 and S14 Figs). Additionally, one, to our knowledge novel, motif, GUCCCC, was found, which significantly associated with repression of MRL in all MPRA datasets. However, the motif GUCCCC did not show a clear association with repressed translation in endogenous datasets (S15, S16, S17, S18 and S19 Figs and S6 Table). Further investigations would be necessary to understand whether and how the motif GUCCCC is a native 5'UTR regulatory element.

To offer an illustration of how variants effects are predicted in a specific sequence, we provide the results of a model interpretation procedure for the 5'UTR sequence of the HBB gene (HBB-001), which plays a role in beta-thalassemia and has known 5'UTR variants [41,42]. Specifically, we computed gradient contribution scores and all predicted variant effects for all positions in the sequence (Fig 7A). Contribution scores measure the gradient of the predicted output with respect to a particular input nucleotide. Contribution scores thus give an indication of the importance of this nucleotide to the final output according to the model (Methods). Variant effect scores meanwhile visualize how the model scores mutations at different points in the sequence. As the HBB gene does not include an uTIS, the model mainly focuses on the sequence at the 5' end and on the sequence preceding the canonical start codon. Most SNVs are predicted to have little impact, with the exception of those creating uAUGs. One of these uAUG-creating mutations is a known pathogenic variant. Introducing it to the sequence greatly changes the model's predictions (Fig 7B), as the main focus of the model now shifts to the created uAUG and its context. Consistent with previous research on the Kozak sequence, the -3 position of the uTIS is predicted to play an important role in modulating the effect of the new start codon [10,14]. The striking difference in variant effect prediction in the reference sequence versus the SNV-containing alternative sequence reflects the high nonlinearity of the model. Rather than just being able to score specific SNVs, the model can also provide predictions for combinations of variants (such as creating an uAUG while simultaneously modifying its -3 position).

Altogether, we believe our model can be used to improve germline or somatic variant interpretation and foresee application of it in rare disease research.

## Discussion

We have introduced frame pooling, a novel neural network operation which performs pooling over distinct frames of an input vector. By combining a convolutional neural network with frame pooling and training on MPRA data, we have created a model which can provide mean ribosome load predictions for 5'UTR of any length. In comparison to the state-of-the-art, our

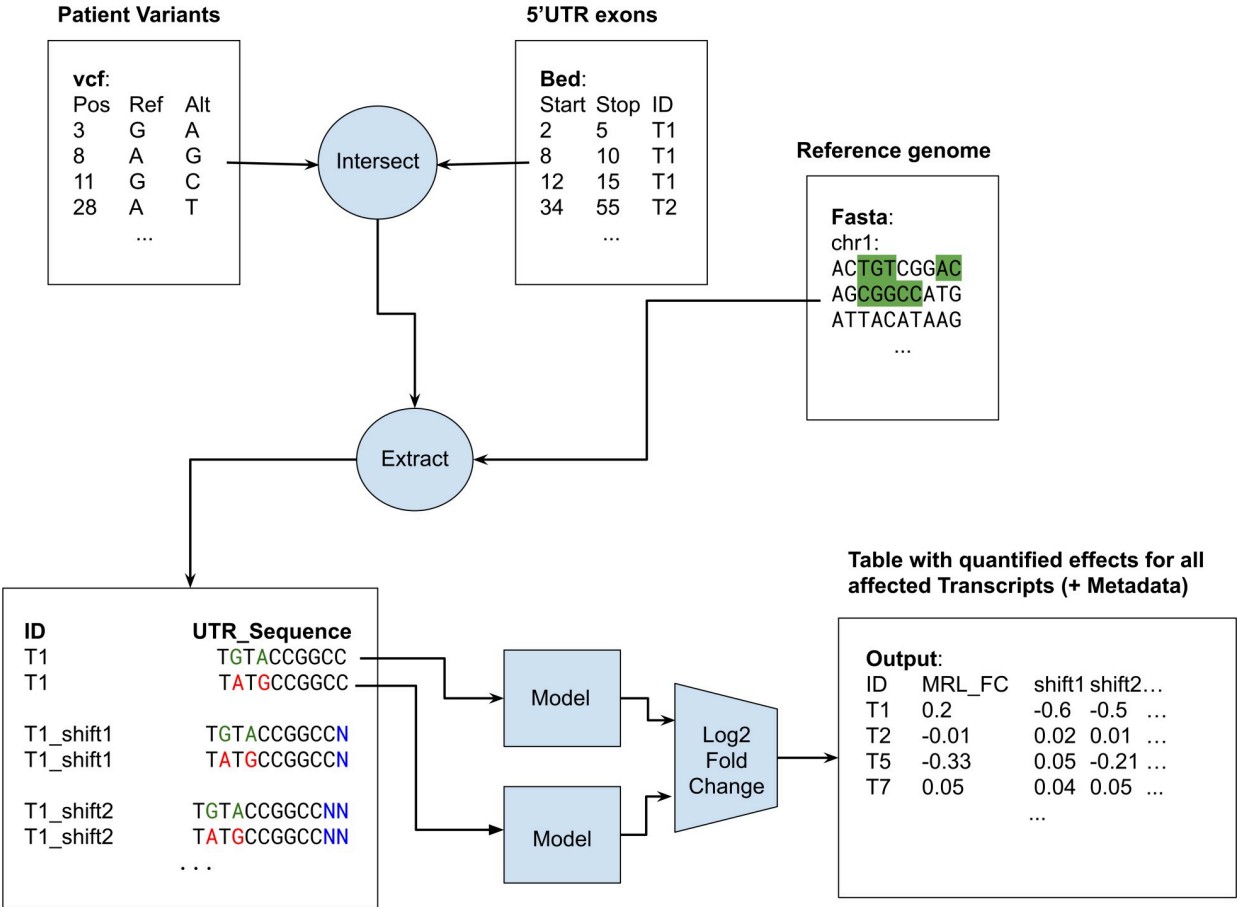

**Fig 6. Schematic of the Kipoi model workflow.** Using a vcf, bed and fasta file as input, the kipoi model predicts log$_2$ fold-change in MRL due to variants affecting particular transcript 5'UTRs. Additionally, the model reports the log$_2$ fold-change in MRL for the same sequences after simulated frameshifts.

approach provides similar performance on short sequences, with the additional advantage that longer sequences do not need to be truncated for analysis. This allows for better generalization and moreover makes it possible to score additional variants which would have been ignored previously. Comparison with conservation scores and previous research on uTIS motifs has further shown that a model equipped with frame pooling has learned to quantify important components of the 5'UTR regulatory code. Since our model has been integrated into the Kipoi framework, it can easily be used by practitioners to analyse any human 5'UTR variant or mutation, including indels. Additionally, it can quickly be integrated into a larger pipeline or serve as the starting point for future research.

The frame pooling approach developed in this paper is specific to the study of translation, but it is reflective of a more general idea: Despite often being regarded as black-box models, it is possible to encode prior biological knowledge into neural networks through careful modifications to their architecture. As demonstrated in this study, this can allow the network to generalize more effectively to unseen contexts and beyond the specific constraints inherent in the training data. Genomics has many features, such as reverse complementarity [43], or helical periodicity which could be encoded in a somewhat analogous fashion into neural network architecture. However, by encoding a specific model of biology into a neural network, one also reduces its ability to learn about mechanisms which deviate from this model.

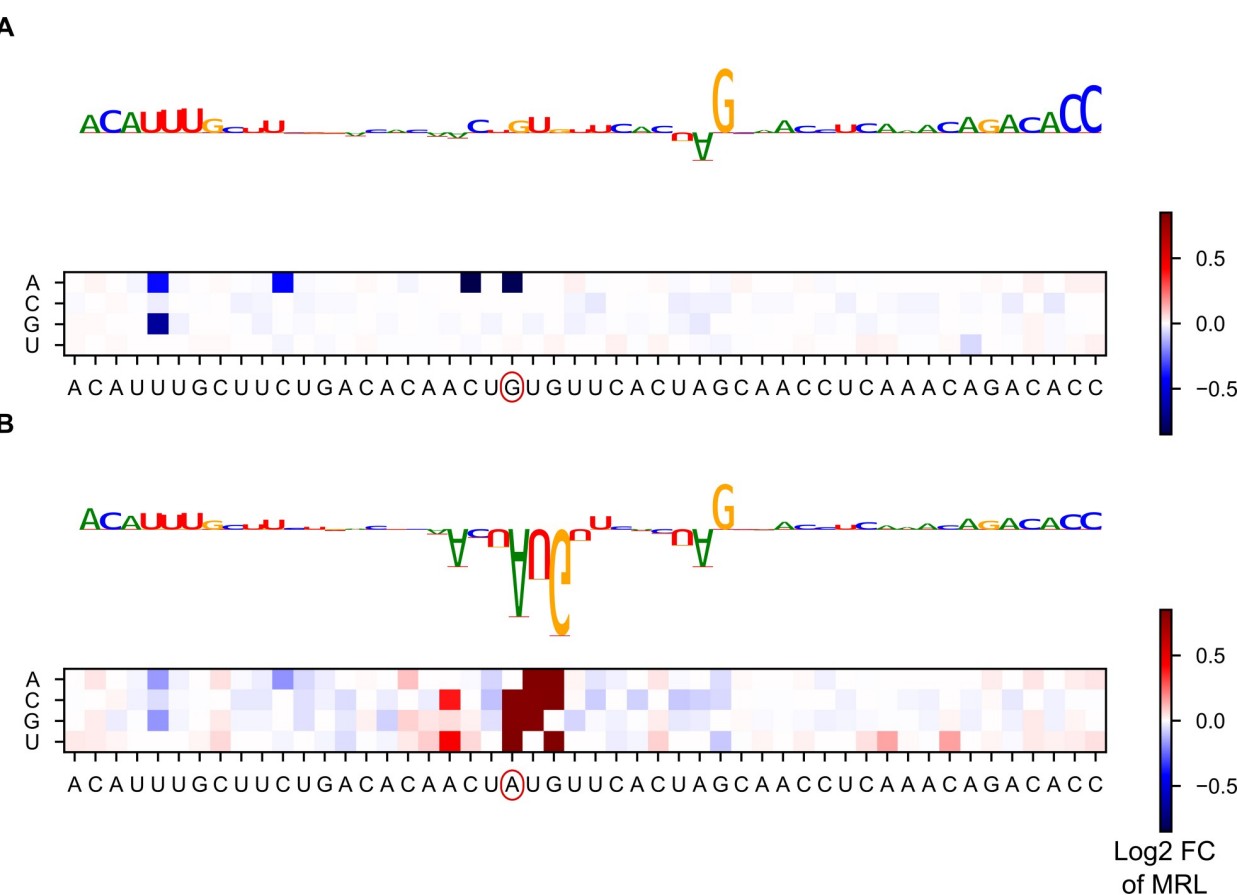

**Fig 7. Identifying motifs within the HBB 5'UTR sequence. A)** The sequence logo shows the gradient impact of each nucleotide on the model's final prediction. The 5' end and the sequence immediately preceding the canonical start contribute relatively strongly to the gradient, as does a G. The heatmap shows the predicted impact (in terms of $\log_2$ fold change) of mutating each position within the sequence. Strong effects are associated with uAUG-creating variants. **B)** Variant rs34704828, a known pathological variant which creates an uAUG, is introduced to the sequence. The model's main focus, in terms of gradient, now lies on the new AUG and its surrounding context. The heatmap shows that the impact of the variant is influenced by its context.

Despite its clear advantages, two limitations of our model architecture should be noted. First, because the architecture relies on convolutions and pooling, there are known features of translational regulation that it cannot fully capture. For one, a frame pooling model cannot detect upstream open reading frames that are longer than the receptive field, that is, the segment of the input sequence that contributes to the activation of an individual neuron in the final convolutional layer. In our architecture, this limits uORF that can reliably be detected to a length of 19 nt. Second, this architecture cannot entirely replicate the mechanics of leaky scanning. While it can reliably detect uTIS, detect which frame they are in and assess their relative strengths, it cannot determine in which order they appear in the sequence. Because the ribosome scans from the 5' end, this ordering should in principle matter, although it is unclear whether this effect is detectable in the MPRA data. A recurrent neural network, such as long short term memory networks [44], could overcome these limitations. However, in our experiments simple long short term memory networks were not able to outperform the frame pooling models and additionally took far longer to train and to predict. Accordingly, designing a scanning model that incorporates our knowledge of the scanning process more holistically, remains an avenue for future research.

More generally, training on MPRA data implies unique advantages and disadvantages. The main advantage of MPRA data is its lack of bias as compared to endogenous data. In endogenous data, many sources of spurious correlations exist as a result of evolutionary optimization. For example, certain classes of genes may be highly translated as they fulfill vital functions. A model may then use motifs in the coding sequence that encode these functions to predict translation. It will then possibly erroneously conclude that mutation of the function also impacts translation. In an MPRA experiment such as Sample et al. [25], such spurious correlations do not arise as everything except the 5'UTR is held constant. A model trained on such data can thus provide unbiased and plausible effects for the impact of a mutation. Moreover, MPRA experiments sample a much larger space of genetic variation than endogenous data. Evolution quickly removes variants which are too destructive from the gene pool, thus collecting sufficient data on highly pathogenic genetic mutations from living populations is difficult. MPRA data circumvents this problem as it allows for any kind of motif to be analysed, no matter if it could occur in a living human. Thus, models trained on MPRA data can more confidently score variants deviating heavily from the average genome. This suggests that our models could be used effectively in a clinical setting to quickly flag candidate mutations that could significantly distort translation.

This lack of bias of randomly designed MPRA dataset however also presents a limitation. Because the space of possible sequences is probed randomly, motifs consisting of many nucleotides are unlikely to be featured in the data, unless the sample size is unfeasibly large. Moreover, motifs with more subtle effects may be drowned out by motifs with larger effects. As a result a model trained on a randomly designed MPRA dataset may capture motifs that rarely occur in living organisms while not capturing complex motifs selected by evolution to finetune biological processes. In particular, up to 10% of human 5'UTR exhibit an internal ribosome entry site (IRES), which allows a ribosome to directly initiate translation without first scanning the 5'UTR [45,46] but IRES motifs did not show up in our model. Making a more general model, which combines knowledge from the Sample et al. [25] MPRA with other studies specific to IRES, presents another avenue for future research. This conclusion extends also to other 5'UTR features, including the overall length of the 5'UTR, 5'UTR introns and factors related to mRNA stabilization and decay, all of which may play a role in regulating translation [18]. Combining MPRA with endogenous data could be an effective method to allow detection of additional motifs, without sacrificing the ability to score variants in an unbiased manner.

## Methods

### Framepool model

All code was written in python 3.6. The model was implemented using Keras with a Tensorflow backend. With the exception of the frame pooling (see below), all layers use the standard Keras API.

The three convolutional layers all have a kernel size of 7 and 128 filters each. The dense layer has 64 neurons and a dropout with a drop-rate of 0.2 is applied to its outputs. All layers, except the final linear layer and the frame pooling, use ReLU activation functions. Altogether the framepool models have 282,625 learnable weights each, with the exception of the FramePoolCombined model, which has 4 additional learnable weights in the scaling regression layer.

Note that residual skip connections [47] were used between convolutional layers in all models. These were used as they speed up training, but did not provide any tangible performance benefits.

Tensorflow requires each batch-tensor to have defined dimensions. To accommodate variable length sequences, we pad the 5' end of all sequences with zeros to match the longest

sequence in the batch. To prevent these additional zeros from impacting predictions, a masking tensor is used. Specifically, this masking tensor ensures that the correct denominator is used in average pooling.

**Frame pooling.**    Frame pooling was implemented using Tensorflow's tensor-slicing utilities. The output tensor of the third convolutional layer is sliced into three tensors, one for each frame with respect to the canonical start codon. Global max pooling and global average pooling are then used to aggregate the filter outputs along the input sequence. The resulting pooled tensors thus indicate whether the motif captured by a particular filter is (a) present somewhere in the sequence and (b) how strongly it is present on average along the sequence. These tensors are concatenated and fed to the dense layer.

To illustrate how this works, consider the following example: a sequence of length 300 (so dimensions are (300, 4) after one-hot encoding) is fed to the model. After the third convolutional layer, it will have dimensions (300, 128). Before slicing, the tensor is reversed along the sequence axis, to ensure that the frame is related to the canonical start codon in a consistent manner for all input sequences. After slicing, there will be three tensors, each with dimensions (100, 128), whereby the first tensor will consist of the slices conv3_output[(0, 3, 6, . . ., 297),:], the second will consist of conv3_output[(1, 4, 7, . . ., 298),:] and so on. Global max pooling will return three tensors of shape (128,), and so will global average pooling. After concatenation, a tensor of shape (768,) remains.

**Model training.**    The FramePool50 model was trained on the egfp1 MPRA dataset from Sample et al. [25]. This set comprises the sequences of 260,000 random 5'UTR, all 50 nucleotides in length, together with mean ribosome load measurements for each sequence.

Of these 260,000 sequences, 20,000 were withheld for the validation set, which was employed to optimize hyperparameters and for early stopping (but not for testing). Hyperparameters were adjusted manually, but generally only minor tweaking was necessary, as the Optimus 5 Prime hyperparameters already worked very well for the framepool model. Training was done using the Adam optimizer with default parameters and a standard mean-squared-error loss function. Training was stopped early if no improvement was made on the validation set for 3 consecutive epochs. This generally happened after 10–15 epochs. The model with the lowest validation set error was then chosen.

The FramePool100 model was trained on the random variable length MPRA dataset. This set comprises 76,319 random 5'UTR, with lengths varying from 25 to 100 nucleotides, together with mean ribosome load measurements for each sequence. No new validation set was constructed for this data and training was done for six epochs straight, without early stopping or other optimizations.

The FramePoolCombined model was trained on both training sets for six epochs straight, with an additional library indicator to account for differences in scaling.

**Model testing.**    Two MPRA test sets were used to evaluate model performance. One consists of 20,000 MPRA sequences that are 50 nt long. The other consists of 7,600 MPRA sequences, which vary in length from 25 to 100 nt, with 100 sequences for each length. These are the same test sets which were used in Sample et al. [25] to evaluate the performance of the Optimus models. Thus, performances on these sets are directly comparable.

**Kipoi integration.**    To make the model compatible with the Kipoi model zoo API, a custom dataloader was written in Python 3.6, using the numpy, pandas and pybedtools packages. The dataloader takes as input a bed file specifying 5'UTR exon regions of interest, a vcf file specifying the variants of interest, and a fasta file with a human reference genome. The dataloader then uses pybedtools to intersect the vcf file with the bed file, keeping only transcripts which have some variant in one of their 5'UTR exon regions. The reference sequence for these

5'UTRs are then extracted from the fasta file, all intersecting variants are injected, and both reference sequences and variant sequences are one-hot encoded and fed to the model in batches.

The model then predicts MRL for both the reference sequence and the variant sequence. Using these predictions, the $\log_2$ fold change of MRL due to the variants is computed and reported.

By zero-padding the 3' end of both reference and variant sequence, a frameshift with respect to the canonical start codon is simulated. The $\log_2$ fold-change of MRL due to the variants for these shifted sequences is reported as additional output, to provide more information to the user. Particularly, this information can be used to detect AUG-creating/destroying variants that act within the canonical frame and thus only lengthen/shorten the canonical protein, rather than destroying it.

**Frame unaware deep learning models.**   To evaluate the benefit of encoding frame information into neural net architecture, we also trained models using conventional global pooling strategies. Specifically, we trained three models, which we call global_pool_conv, dilated_conv and more_dilated_conv, respectively. The first, global_pool_conv used the same hyperparameters as FramePool50, but the frame pooling layer was replaced with conventional global max and average pooling. This allows the model to predict on arbitrary length sequences, but provides no explicit frame information.

Dilated_conv uses the same architecture as the global_pool_conv, but the second and third convolutional layers are dilated. Specifically, the second convolutional layer has a dilation of 2, and the third a dilation of 4. This gives the model a receptive field size of 43, which almost spans the 50nt long sequences. More_dilated_conv adds a fourth convolutional layer, with a dilation of 8.

All these models were trained on the same training set as FramePool50.

**Random forest models.**   To evaluate whether a deep learning approach outperforms other machine learning approaches, we also trained random forest models which use k-mer counts as input features. By representing the input sequence as a fixed-length vector of k-mer counts, we ensure that our random forests can, in principle, handle sequences of any length.

We trained two such random forest models: 3mer_random_forest and 4mer_random_forest, which use 3-mer and 4-mer counts as input features, respectively. The random forests were trained using scikit-learn [48], with default hyperparameters (100 trees, no maximum depth, all features used at every split), on the same training set as FramePool50.

To evaluate the impact of providing random forests with frame information, we additionally trained what we call the framed_forest model. This random forest takes as input the k-mer-counts for each frame (giving a vector of length $3^*(4^k)$). We trained the framed_rf model using the same hyperparameters as the 4mer_rf and on the same training set.

We additionally performed a Bayesian hyperparameter optimization for the framed random forest using the tree of Parzen estimators algorithm from the hyperopt [49] package. We found that the optimal hyperparameters were 200 trees with a max depth of 32, however this essentially had the same performance as using the defaults. We also trained a framed random forest on 5-mers, to see if increasing the k-mer size improved results, but it did not.

## TIS strength

Noderer et al. [9] provide estimates for the relative translation initiation strength of all possible -6 to +5 contexts surrounding an AUG start codon. This data was downloaded from the journal website.

The frame pooling models only output MRL predictions and thus do not rate the strength of different TIS contexts directly (although internally they undoubtedly have some notion of what constitutes a strong TIS). To force the models to provide TIS-strength predictions, we

took each AUG and their context of the Noderer et al data, added random bases on either side (so the sequence length equals the receptive field size of the convolutional layers), and then predicted MRL twice: once with AUG activated, and once with an inactive start (AGG). Then the predicted fold change in MRL due to AUG activation was correlated with the TIS Strength as estimated by Noderer et al [9]. To reduce noise due to the added random bases, this procedure was performed 100 times and an average was taken. Additionally, contexts which inadvertently introduce another AUG, e.g. AAUGGGAUGGG, were removed from the analysis, as in such a case it is not clear which of the two AUG is selected as start codon. Accordingly, the surrounding context of the start codon is not well defined.

The same analysis was performed for each non-AUG start codon considered in Diaz de Arce et al [31]. Again, contexts which inadvertently introduce an AUG were removed.

### Translation-related measures of endogenous genes

To evaluate th predictive performance on endogenous sequences, data from a variety of experiments used to investigate translational control were collected. These include a Trip-Seq experiment [34], three Ribo-Seq experiments [35–37], and two protein-to-mRNA ratio (PTR) experiments [3,38].

For the Trip-Seq experiment, processed data at the transcript level was downloaded from the journal website. Transcripts with a count less than 1 transcript per million (TPM) in either replicate were filtered and then the replicates were averaged. Next, counts for transcripts with the same 5' UTR were aggregated (as all models considered here only focus on the 5' UTR) and the mean ribosome load was calculated from the polysome fractions. The final dataset has MRL values for 25,831 transcripts.

For the Ribo-Seq experiments, processed data at the gene level was downloaded from the respective journal websites. In each case, only data from the control condition was used, as the model was not trained to predict translation under stress or other abnormal conditions. The Andreev and Xiao datasets were filtered, to exclude sequences with less than 10 Ribo-Seq reads, as these produced outliers. To compute a measure similar to MRL which can be correlated with the model predictions, the RPF (ribosome protected fragment) read count was divided by the RNA-seq read count for each sequence, yielding the Ribo-Seq ribosome load, also called TE (Translation Efficiency). This procedure yielded TE values for 8003, 7672 and 8956 genes in the Andreev, Xiao and Eichhorn datasets respectively.

For the PTR experiments, processed data for tissue-specific major transcripts were downloaded from the respective journal websites. As the models considered here are not tissue-specific, the median PTR across tissues was calculated for each available transcript. This procedure yielded PTR values for 5,293 transcripts in the Wilhelm dataset and for 11,575 transcripts in the Eraslan dataset.

For each dataset, the respective sequences were then one-hot encoded and fed to FramPool50/100 (not truncated) and Optimus50/100 (after truncation to the required fixed-size). Then Pearson and Spearman correlations between the model predictions and the observed measures (MRL, TE or PTR) were computed.

To evaluate whether the performance difference between models was significant, the following procedure was used: first, the difference in performance between the models on 100 bootstrap samples was computed. From this, we calculated the standard deviation of this difference. Note: we use standard deviations here, and not standard errors, as standard errors can be made arbitrarily small simply by increasing the number of bootstrap samples. This is undesirable, as additional bootstrap samples likely add far less information than additional samples from the actual population.

Using the standard deviation, a 95% confidence interval was constructed for the difference. Since 6x4 = 24 tests need to be conducted to perform all comparisons, a Bonferroni correction was applied. Thus, the constructed intervals for individual data points correspond to 99.98% CI, which is equivalent to +/-3.54 standard deviations on the normal distribution. A difference is considered significant if the CI does not overlap zero.

## Predicting variant effects for LoF-intolerant genes

The predicted effect of every possible 5'UTR single nucleotide variant in the canonical transcript of every loss-of-function intolerant gene was computed. To define a loss-of-function intolerant gene, we use the observed/expected (oe) score as provided by gnomAD (gnomAD version 2.1, [50]). Specifically, any gene where the upper bound of the 90% oe score lies below 0.35 is classed as loss-of-function intolerant, which is the cutoff recommended by the gnomAD consortium. The oe score compares the number of loss-of-function variants observed in a gene with the number that would be expected to arise by chance given a mutational model that considers sequence context, coverage and methylation. The data was taken from the "pLOF by Transcript TSV" table from the gnomAD website, which also defines which transcripts were considered as canonical for the purpose of computing the oe score.

For possible SNVs located at most 100 nt upstream of the start codon of the respective transcript, the sequence was truncated, and the SNV effect was computed both using Frame-Pool100 and Optimus100, to allow for a fair comparison. For those located further than 100 nt from the start, only the FramePool100 model can be used.

As phyloP conservation scores do not distinguish between variant bases, the average variant effect was computed by averaging the effect of the three possible SNV at each position (note that we average the absolute values of the $\log_2$ fold changes, to ensure up and downregulation are treated uniformly). This gives an indication of how impactful a mutation is at a particular position in expectation, without knowing to which base it will mutate. Moreover, since a particular position could be included in the 5'UTR of more than one transcript, the variant effect of each position was also averaged across all 5'UTRs that contain it.

PhyloP conservation scores (hg19 phyloP 100way) were downloaded in bigwig format from the UCSC website. Positions in the genome were binned according to the model's predicted average variant effect and the average phyloP score for each bin was computed. Additionally, error bars corresponding to a 95% t-test confidence interval were calculated.

## Contribution scores

Using the DeepExplain package [51], the gradient times input metric was computed for a variety of sequences. Gradient times input, as the name implies, computes the gradient for a particular input example (that is, a particular 5'UTR sequence) and then multiplies it with the input tensor. The resulting contribution scores measure the impact of slightly strengthening a base on the predicted MRL. Mathematically, as the bases are one-hot encoded, this means that the score measures the impact on the output of infinitesimally increasing the one-hot value for a base from the value 1. While this does not have a direct biological interpretation (there is no such thing as infinitesimally more than one Uridine), it nevertheless indicates which bases the model deems particularly important for its prediction (since, if a base is not important for the prediction, perturbing it should not influence the output).

Using the concise package [52] these contribution scores are then visualized as sequence logo. This allows for easy interpretation.

## Tf-Modisco motif search

Tf-Modisco [40] is a method to "obtain a non-redundant set of predictive motifs learned by the neural network". The algorithm takes as input per-base contribution scores and then uses a clustering strategy to find important motifs. This method yields more interpretable results than individually visualizing convolutional filters, as such filters often learn distributed representations of sequence features. More detail can be found in the respective paper and in Shrikumar et al. [53] and Avsec et al. [54].

To run Tf-Modisco, we computed gradient times input contribution scores for all length 50 MPRA sequences. We additionally computed the raw gradient contribution scores for the same sequences, to serve as hypothetical contribution scores, as explained in the Tf-Modisco paper [40]. We then ran Tf-Modisco using a sliding window size of 10, a flank size of 5, a trim to window size of 15 and the default Laplacian null distribution with an FDR cutoff of 0.15.

We additionally repeated the same analysis using DeepLift [55] contribution scores, but found the same motifs.

## Supporting information

**S1 Fig. Performance of Optimus50 and Framepool50 on (truncated) human 5'UTR sequences measured in the MPRA experiment.** On 50 nt human sequences, the two models perform equivalently, but FramePool50 generalizes much better to longer sequences.
(TIF)

**S2 Fig. Performance of Optimus100 and Framepool100 on fixed and variable length MPRA sequences.**
(TIF)

**S3 Fig. Performance of Optimus100 and FramepoolCombined on fixed and variable length MPRA sequences.**
(TIF)

**S4 Fig. Performance of gobal_pool_conv and FramePool50 on fixed and variable length MPRA test sequences, when trained on fixed length sequences.** The global_pool_conv uses the same hyperparameters as FramePool50, but frame pooling is replaced with standard global max and average pooling. The global_pool_conv, which has no knowledge of frame, performs worse than FramePool50.
(TIF)

**S5 Fig. Performance of dilated_conv and FramePool50 on fixed and variable length MPRA test sequences, when trained on fixed length sequences.** The dilated_conv uses the same hyperparameters as FramePool50, but frame pooling is replaced with standard global max and average pooling and, additionally, dilations are used to expand the receptive field size to 43. On fixed length data, the large receptive field size likely allows the dilated_conv model to effectively infer the frame, but this fails to fully generalize to the variable length sequences. As a result, FramePool50 generalizes markedly better to the variable length sequences.
(TIF)

**S6 Fig. Performance of more_dilated_conv and FramePool50 on fixed and variable length MPRA test sequences, when trained on fixed length sequences.** The more_dilated_conv model is similar to the dilated_conv model, but has one additional dilated convolutional layer (with a dilation factor of 8). The additional dilation brings no improvement.
(TIF)

**S7 Fig. Performance of 4mer_random_forest and frame_forest on fixed and variable length MPRA test sequences, when trained on fixed length sequences.** The 4mer_random_-forest gets as input a 4^4-dimensional vector of 4-mer counts, which offers no frame information. The frame_forest gets as input a 3*(4^4)-dimensional vector of 4-mer counts for each frame. The frame information allows the frame_forest model to markedly outperform the frame-unaware random forest.
(TIF)

**S8 Fig. Performance of frame_forest and FramePool50 on fixed and variable length MPRA test sequences, when trained on fixed length sequences.** FramePool50 offers better performance than the random forest based model.
(TIF)

**S9 Fig. Correlation of predicted and measured uTIS strengths for FramePool50.**
(TIF)

**S10 Fig. Correlation of predicted and measured uTIS strengths for FramePool100.**
(TIF)

**S11 Fig. Contribution scores for a synthetic sequence, before and after a GUCCCC motif is inserted.** Contribution logos are aligned for visualization purposes. Since a frame pooling model can predict MRL for sequences of any length, it can in principle quantify the effect of any indel or complex variant. This is done by predicting MRL for both the wildtype and the mutated sequence and then computing the log-fold change in MRL. In this case, the insertion of the GUCCCC motif is predicted to depress MRL.
(TIF)

**S12 Fig. Contribution scores for a synthetic sequence, before and after a C is mutated to a G, creating an in-frame uTIS.** Because the new uTIS is in-frame with respect to the canonical start codon, it has only a small effect on the MRL predicted by the model. However, such a variant could be lengthening the canonical protein. To alert the user to such in-frame uTIS creation or deletion events, the kipoi version of the model also reports the effect of the variant after shifting the frame of reference. This shift "tricks" the model into treating an in-frame variant as out-of-frame. If the variant effect on MRL after such a shift is strongly negative, whereas the original effect is small, it is an indication that a variant may be lengthening the canonical protein.
(TIF)

**S13 Fig. uTIS motifs found by Tf-Modisco.**
(TIF)

**S14 Fig. Stop codon motifs found by Tf-Modisco.**
(TIF)

**S15 Fig. The GUCCCC motif found by Tf-Modisco.**
(TIF)

**S16 Fig. The effect of the GUCCCC motif in the different MPRA datasets.** In each dataset, presence of the GUCCCC motif leads to a repression of mean ribosome load compared to when the motif is not present.
(TIF)

**S17 Fig. The effect of the presence of the GUCCCC motif in endogenous genes in the Trip-Seq dataset.**
(TIF)

**S18 Fig. The effect of the presence of the GUCCCC motif in endogenous genes in the ribosome profiling datasets.** TE is a measure of translation efficiency given by the log-ratio of ribo-seq reads to rna-seq reads covering a specific gene.
(TIF)

**S19 Fig. The effect of the presence of the GUCCCC motif in endogenous genes in the Eraslan et al. dataset.** PTR is the protein-to-RNA ratio.
(TIF)

**S1 Table. Performance on MPRA datasets of Optimus and Framepool models.**
(XLSX)

**S2 Table. Pearson correlations of predictions with endogenous data.** * = p-value smaller than $1*10^{-10}$, ** = p-value smaller than $1*10^{-50}$ (where $H_0$ is zero correlation).
(XLSX)

**S3 Table. Spearman correlations of predictions with endogenous data.** * = p-value smaller than $1*10^{-10}$, ** = p-value smaller than $1*10^{-50}$ (where $H_0$ is zero correlation).
(XLSX)

**S4 Table. Performance on MPRA datasets of different random forest models.** The frame_forest_hyperopt model uses the optimized hyperparameters which were found using the hyperopt package.
(XLSX)

**S5 Table. Results of a linear regression to predict MRL in the fixed length MPRA dataset.** We see that the GUCCCC motif has a statistically significant effect on MRL even when controlling for GC content and (predicted) UTR min folding energy.
(PNG)

**S6 Table. Results of Wilcoxon rank-sum tests comparing genes which have the GUCCCC motif at least once to those that do not.**
(XLSX)

## Acknowledgments

We would like to thank the Seelig lab, particularly Ban Wang, for providing their code and data.

## Author Contributions

**Conceptualization:** Alexander Karollus, Žiga Avsec, Julien Gagneur.

**Formal analysis:** Alexander Karollus.

**Investigation:** Alexander Karollus.

**Methodology:** Alexander Karollus.

**Project administration:** Julien Gagneur.

**Resources:** Julien Gagneur.

**Software:** Alexander Karollus.

**Supervision:** Žiga Avsec, Julien Gagneur.

**Validation:** Alexander Karollus, Žiga Avsec.

**Visualization:** Alexander Karollus.

**Writing – original draft:** Alexander Karollus.

**Writing – review & editing:** Julien Gagneur.

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
