## [Decision Letter · Decision Letter 0]

25 Jan 2021

Dear Mr Karollus,

Thank you very much for submitting your manuscript "Predicting Mean Ribosome Load for 5’UTR of any length using Deep Learning" for consideration at PLOS Computational Biology.

Let me first apologize for the extraordinary time it took to review your article. As you might imagine, the significant load on reviewers and their reluctance to agree on reviews or complete the reviews they agreed to do is the main reason behind this delay. However, I am glad that we obtained two reviews that will allow us to see your paper in the next stage. We have one outstanding review that may still come but in the interest of moving the science forward, we have decided to not wait for it at this time.

As with all papers reviewed by the journal, your manuscript was reviewed by members of the editorial board and by independent reviewers. In light of the reviews (below this email), we would like to invite the resubmission of a significantly-revised version that takes into account the reviewers' comments.

We cannot make any decision about publication until we have seen the revised manuscript and your response to the reviewers' comments. Your revised manuscript is also likely to be sent to reviewers for further evaluation.

Sincerely,

Predrag Radivojac

Associate Editor

PLOS Computational Biology

William Noble

Deputy Editor

PLOS Computational Biology

Reviewer's Responses to Questions

**Comments to the Authors:**

Reviewer #1: Review is uploaded as an attachment.

Reviewer #2: The authors predict the mean ribosome load from 5’UTR sequences. Overall, the paper is well written, and has a sound methodology towards predicting the translation rate. The main contribution of the paper is the framepooling operation within a neural network where a convolutional neural network takes into account the three possible frames individually by taking the global average and max pooling from each of them. This allows the model to learn features from arbitrary length sequences by having full coverage of the sequences.

Pros:

1. The model performs better or at a similar level of the Optimus models. Specially it performs really well in terms of correlation score for longer than 100nt sequences.

2. The frame pooling operation is novel in the context of this problem.

3. The model performs fairly well in predicting strength of upstream transcription initiation sites.

4. The model achieves fairly well PhyloP conservation scores from its prediction of single nucleotide variants showing it may indeed be learning biologically relevant subsequences.

5. The authors tested their method on endogenous data as MPRA data might not account for the spuriousness that is experienced there. Even though the correlation scores are pretty low, I thank the authors for undertaking this part.

Cons:

1. The absence of a convolutional neural network that takes arbitrary length sequences and uses traditional dilated convolution or max pooling operations mean I am not really sure the frame pooling operation is indeed necessary.

2. No comparison with a random forest that just takes as input 3-grams or 4-grams. This solves the arbitrary length issue, and from experience, it’s hard to do better than a random forest. So, it is essential that there is a comparison to gauge if a neural network is even necessary.

3. One salient point of using a neural network might be to interpret features learnt by the convolutional filters. It will be interesting to see if they are learning motifs but there is not much material on this in the paper.

The paper does a good job of presenting the results in a clear way. The authors have also deposited their code and data properly which is always appreciated. I thank the authors for their hard work and would like to see their responses to my cons comments.

**Have all data underlying the figures and results presented in the manuscript been provided?**

Reviewer #1: Yes

Reviewer #2: Yes

PLOS authors have the option to publish the peer review history of their article (what does this mean?). If published, this will include your full peer review and any attached files.

Reviewer #1: **Yes: **Anthony D. Fischer

Reviewer #2: No
---

## [Decision Letter · Decision Letter 1]

19 Apr 2021

Dear Mr Karollus,

We are pleased to inform you that your manuscript 'Predicting Mean Ribosome Load for 5’UTR of any length using Deep Learning' has been provisionally accepted for publication in PLOS Computational Biology.

Best regards,

Predrag Radivojac

Associate Editor

PLOS Computational Biology

William Noble

Deputy Editor

PLOS Computational Biology

Reviewer's Responses to Questions

**Comments to the Authors:**

Reviewer #2: I thank the authors for addressing the questions/concerns I had.

**Have the authors made all data and (if applicable) computational code underlying the findings in their manuscript fully available?**

Reviewer #2: Yes

PLOS authors have the option to publish the peer review history of their article (what does this mean?). If published, this will include your full peer review and any attached files.

Reviewer #2: No

---

## [Editor Report · Acceptance letter]

5 May 2021

PCOMPBIOL-D-20-01257R1 

Predicting Mean Ribosome Load for 5’UTR of any length using Deep Learning

Dear Dr Karollus,

I am pleased to inform you that your manuscript has been formally accepted for publication in PLOS Computational Biology. Your manuscript is now with our production department and you will be notified of the publication date in due course.

With kind regards,

Katalin Szabo
